# StarCraft II Arena: Evaluating LLMs in Strategic Planning, Real-Time Decision Making, and Adaptability

## Abstract

StarCraft II plays an important role in developing AI agents for real-time strategic reasoning due to its complex nature. However, people usually draw conclusions of how competent their agents are according to the level of the built-in agents in Star-Craft II which they can win in terms of the final success rate. Little intermediate quantitative information is considered while human-in-the-loop analysis is time inefficient, which results in inadequate reflection of the true strategic reasoning ability. In this work, we propose StarCraft II Arena, a well-designed benchmark for evaluating the strategic planning, real-time decision-making, and adaptability capabilities of large language models (LLMs) agents. We introduce using fine-grained capability metrics, allowing for targeted capture and analysis of specific capability, and further propose a detailed decision trace to enhance the under-standing of LLM behavior. We demonstrate the utility of such a benchmark by evaluating several state-of-the-art LLMs in various setups. Our results reveal distinct performances in long-term strategy development, real-time decision-making, and adapting to environmental changes. Such results show that the StarCraft II Arena offers a deeper insight into the decision-making process of LLMs and has the potential to become a challenging and comprehensive benchmark for strategic reasoning.

## 1 Introduction

LLMs have recently demonstrated exceptional capabilities in reasoning, planning, and problem-solving (Xi et al., 2023) across a range of domains, such as policy formulation (Xiao et al., 2023; Hua et al., 2023), investment decision-making (Weiss et al.; Li et al., 2023b), and strategic optimi-sation (Liu et al., 2024; Zhang et al., 2024a). Successfully completing these complex tasks requires intelligent agents to perceive, make decisions, and execute actions (Wooldridge & Jennings, 1995) within diverse and dynamic environments. This process not only involves deep reasoning to antic-ipate risks and weaknesses but also the ability to understand the motivations, beliefs, and potential deceptive behaviors of other agents (Hao et al., 2023; Premack & Woodruff, 1978; Street et al., 2024). Although LLMs have shown significant promise in managing such scenarios, positioning them as key technologies for achieving artificial general intelligence (AGI) (You et al., 2024; Morris et al.), their performance in real-world applications continues to face numerous challenges.

Evaluating the capabilities of LLM agents effectively is critical for the further development of this field. Traditional static evaluation datasets, while offering a standardized testing framework (Wang et al., 2019; Srivastava et al., 2022; Chen et al., 2021; Xie et al., 2024), are insufficient for captur-ing how models make decisions and adapt in dynamic environments. As a result, there has been growing interest in assessing the performance of large models within executable environments (Liu et al., 2023a; Xi et al., 2024)—simulated or real-world interactive platforms, including web naviga-tion (Lai et al., 2024), household tasks (Li et al., 2024), gaming (Bailis et al., 2024; Qi et al., 2024), and programming (Qian et al., 2024). Among these, games, with their clear rules and complex decision-making mechanisms, are considered ideal platforms for evaluating AI decision-making abilities (Costarelli et al., 2024).

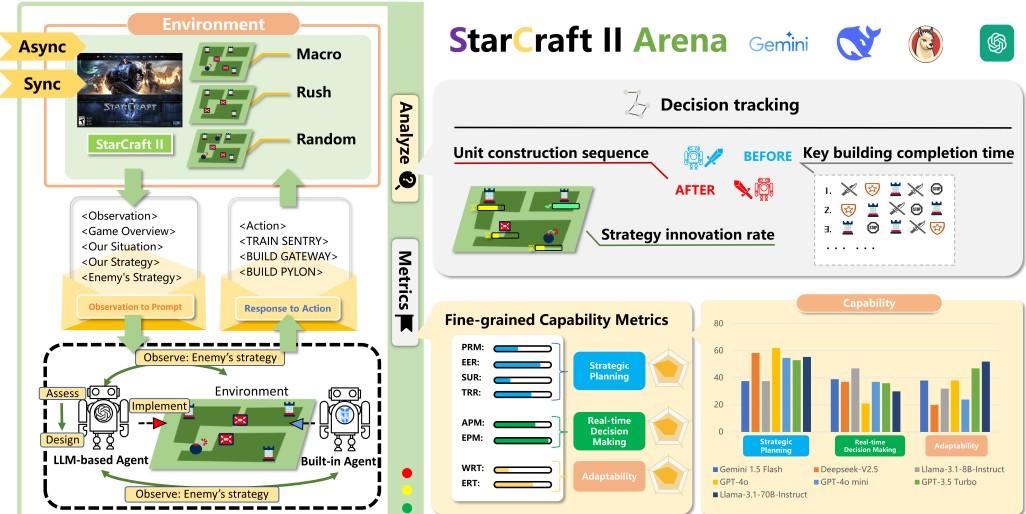

Figure 1: The overall framework of **StarCraft II Arena** which is designed to evaluate LLMs in strategic planning, real-time decision-making, and adaptability. It uses fine-grained capability metrics and a decision tracking system to capture key elements like Unit Construction Sequence and Strategy Innovation Rate, providing insights into LLMs' decision-making and strategic reasoning.

However, existing evaluation benchmarks usually take the final outcome of a game such as success rate as the primary measure of performance (Duan et al., 2024). They further neglect the details of the intermediate outcomes gradually generated by LLM agents during the sequential decision-making process (Xi et al., 2024; Ma et al., 2024). Although the success rate reflects the overall ability of the competing agents, this singular metric is usually inadequate for reflecting how capable an LLM agent is when handling complexity or adapting to changing tasks. Therefore, a more comprehensive evaluation approach is required to better reflect the actual underlying reasoning capabilities of LLMs in dynamic environments.

In this study, we introduce **StarCraft II Arena**, a benchmark specifically designed to evaluate the abilities of LLMs for strategic planning, real-time decision-making, and adaptation in the game StarCraft II. As illustrated in Figure 1, unlike traditional benchmarks which usually depend on static tests or success rates, StarCraft II Arena incorporates fine-grained capability evaluation metrics, which allows for more detailed analysis of the performance of LLMs across multiple dimensions. Moreover, we also introduce a decision-tracking mechanism which records the intermediate decision results of LLMs during the task execution. By looking into the decision trajectories, we can analyze how LLMs adjust strategies in response to dynamic changes in the environment. As a result, it allows a more comprehensive understanding of the underlying decision process of LLMs rather than using the final outcome only. We demonstrate the utility of StarCraft II Arena by applying it to a range of recent LLM agents, both proprietary and open-source, and leading to the following key findings: (1) proprietary LLMs excel in long-term strategic planning and resource management but demonstrate limitations in dynamic environments that require rapid adaptation; (2) most existing LLMs struggle with handling incomplete information and adapting to rapidly evolving opponent strategies, limiting their ability to respond effectively to shifting tactics; and (3) smaller models show greater flexibility in real-time decision-making, particularly in high-frequency decision-making tasks, where they often outperform their larger counterparts. These findings highlight the potential of StarCraft II Arena as a challenging benchmark for LLMs agents in strategic reasoning tasks.

## 2 RELATED WORK

### 2.1 LLM-AS-AGENT

The application of large language models (LLMs) as agents is rapidly evolving, encompassing a diverse range of scenarios from single-agent to multi-agent systems (Xi et al., 2023). Early reinforcement learning (RL) agents learned through trial and error in complex environments, but they

| Game | Imperfect Information | Strategic & Tactical | Dynamic space | Real-time v.s. Turn-based |
|------|:---:|:---:|:---:|:---:|
| Civilization(Wikipedia, 2024a) | ✔ | ✘ | ✔ | Turn-based |
| Dota 2(Wikipedia, 2024f) | ✔ | ✘ | ✔ | Real-time |
| Honor of Kings(Wikipedia, 2024c) | ✔ | ✘ | ✔ | Real-time |
| Diplomacy(Wikipedia, 2024b) | ✘ | ✘ | ✘ | Turn-based |
| WerewolfWikipedia (2024d) | ✘ | ✘ | ✘ | Turn-based |
| **StarCraft II**(Wikipedia, 2024e) | ✔ | ✔ | ✔ | Real-time |

Table 1: Compare several games as LLM benchmarking environments based on four key dimensions: Imperfect Information, Strategic & Tactical, Dynamic Space, and Real-time v.s. Turn-based. Games like StarCraft II and Dota 2 feature imperfect information and dynamic spaces, which present significant challenges to the decision-making capabilities of LLMs. StarCraft II uniquely integrates both strategic and tactical elements, making it particularly suitable as a benchmark for assessing LLMs' planning and decision-making abilities. Turn-based games like Civilization and Diplomacy provide a more controlled environment for long-term strategic planning, while real-time games impose time constraints that test the models' ability to react swiftly.

were typically suited only for highly structured tasks and required substantial training time and data (Pourchot & Sigaud, 2018). In contrast, LLM-based agents, trained on extensive text datasets, possess strong language understanding, instruction-following, and generation capabilities (Liu et al., 2022; Lu et al., 2023), enabling them to flexibly navigate varied situations and demonstrate few-shot and zero-shot generalization abilities (Wei et al., 2021; Yao et al., 2022), thus achieving seamless task transfer. Furthermore, these LLM agents exhibit advanced cognitive abilities akin to human intelligence, including chain-of-thought reasoning (Wei et al., 2022; Jin & Lu, 2023; Zhang et al., 2023), planning (Huang et al., 2024a), self-reflection (Madaan et al., 2024), memory (Zheng et al., 2023a; Zhang et al., 2024b), and learning (Zhang et al., 2024a; Xi et al., 2024). These capabilities empower LLM agents to effectively tackle complex decision-making scenarios.

In multi-agent systems, LLMs must not only interact with their environment but also engage in effective communication and collaboration among multiple agents to accomplish tasks (Pourchot & Sigaud, 2018). Such systems emphasize the importance of agent communication and cooperation, allowing them to operate within dynamic and complex environments, such as game simulations (Xu et al., 2023), financial market analysis (Chen et al., 2023), and software development (Qian et al., 2024). Strategic reasoning is particularly crucial in this context, as it requires agents to understand and predict the actions of other agents and adjust their strategies accordingly.

## 2.2 BENCHMARKS FOR AI AGENTS

**Evaluation Environments.** In previous research, the capabilities of large models have primarily been assessed through the construction of static datasets (Wang et al., 2019; Srivastava et al., 2023; Zheng et al., 2023b; Yue et al., 2024; Xie et al., 2024). While an increasing number of benchmarks have introduced broader tasks and datasets, most remain confined to traditional tasks and fail to comprehensively evaluate the capabilities of large language models (LLMs) in open-ended generation, multi-turn interactions, and agent-based roles (Gur et al., 2023; Huang et al., 2024b; Liu et al., 2023b). As LLMs become more adept at addressing real-world challenges, there is a growing trend towards evaluation methods that are based on executable environments rather than static datasets (Gur et al., 2023; Wang et al., 2023; Shinn et al., 2024). Specifically, researchers are now focusing on areas such as web navigation (Deng et al., 2024; Yao et al., preprint), text-based games (Bailis et al., 2024; Mukobi et al., 2023), household tasks (Wang et al., 2022), digital games (Qi et al., 2024; Ma et al., 2023), avatar tasks (Han et al., 2024), tool usage (Tang et al., 2023), and programming (Qian et al., 2024; Zheng et al., 2023b), all of which provide a more realistic context for assessing LLMs. In particular, games are widely regarded as ideal experimental platforms for evaluating the decision-making capabilities of large models (Liu et al., 2023a). By placing models in dynamic and complex gaming environments, researchers can effectively gauge their performance in real-world scenarios.

With clear rules and flexible customization, **gaming environments** have been applied widely for evaluating AI decision-making abilities. We compare several popular games as LLM benchmarking environments in Table 1 based on four key dimensions: Imperfect Information, Strategic & Tactical, Dynamic Space, and Real-time v.s. Turn-based. Notably, StarCraft II, as a complex real-time strategy game (Vinyals et al., 2019; Samvelyan et al., 2019), provides an ideal platform for evaluating LLMs' capabilities in strategic reasoning and multi-agent interaction. By assessing LLM performance in this environment, we can gain deeper insights into how these models respond to complex decision-making and dynamic changes.

**Evaluation Metrics.** Some studies employ game-theoretic tools to systematically evaluate the decision-making abilities of large models within games, aiming to measure their strategic choices and adaptability (Duan et al., 2024). However, these studies often concentrate on simple games with a single dimension, failing to fully capture the complexity of the models' decision-making processes. Additionally, other research has focused on dissecting the capabilities of large models to explore performance variations and potential advantages across different gaming scenarios (Wu et al., 2023; Ma et al., 2024). Nevertheless, most analyses predominantly emphasize win rates, lacking fine-grained capability metrics and decision trajectory analyses, which limits a comprehensive understanding of the models' performance (Costarelli et al., 2024; Duan et al., 2024; Liu et al., 2023a; Wu et al., 2023).

## 3 PRELIMINARY

The agent's interaction with the environment in StarCraft II is modeled as a Partially Observable Markov Decision Process (POMDP), defined by the tuple $\langle W, S, A, O, T \rangle$, where $W$ represents the victory goal, $S$ is the state space, $A$ is the valid actions space, $O$ is the observation space (including environmental feedback), and $T$ is the state transition function. The agent interacts with the environment by selecting actions from $A$ based on the current state $S$ and observations $O$, with the state evolving according to $T$.

**Two-level inference.** In StarCraft II, the complexity of reasoning arises from the need to handle a large observation space and multi-dimensional strategic tasks. This requires two levels of reasoning: high-level strategic planning, such as resource management and army mobilization, and low-level decision-making, such as micro-control in local battles.

$$p_\pi(\tau) = p(s_0) \prod_{t=0}^{T-1} p(a_t^{high}|s_t, c^{high}) \cdot p(a_t^{low}|s_t, a_t^{high}, c^{low}) \cdot T(s_{t+1}|s_t, a_t^{low}, f_t) \quad (1)$$

Here, $a_t^{high}$ represents a high-level decision based on the global strategy $c^{high}$, and $a_t^{low}$ is a low-level action based on local feedback $c^{low}$. The state transition function $T(s_{t+1}|s_t, a_t^{low}, f_t)$ models how the environment transitions in response to the agent's low-level actions and the feedback received from the environment.

### 3.1 FINE-GRAINED CAPABILITY METRICS

To evaluate the specific capabilities of the Large Language Model (LLM), each model is tested in $m$ scenarios, and performance is assessed based on aggregated metrics. A final capability score is calculated as follows:

$$T = \sum_{i=1}^{m} W_i \cdot \beta_i \cdot \left( \frac{1}{n} \sum_{j=1}^{n} \frac{\overline{R}_{y_j} - \mu_j}{\sigma_j} \right) \quad (2)$$

Here, $W_i$ represents the weight of scenario $i$, and $\beta_i$ is a moderating factor for scenario $i$. For each metric $j$, $\overline{R}_{y_j}$ is the average result across $k_j$ runs, $\mu_j$ and $\sigma_j$ are the mean and standard deviation of the metric, used for normalization. The final score is a weighted sum of normalized metrics across all scenarios.

| Capacity | Metrics | Scene selection | |
| --- | --- | --- | --- |
| | | Opponent strategy | Operation mode |
| Strategic Planning | RPM, EER, SUR, TRR | Macro | Async/Sync |
| Real-time Decision Making | APM, EPM | Rush | Async/Sync |
| Adaptability | WRT, ERT | Random | Async/Sync |

Table 2: Outlines the capacity and metrics utilized in the StarCraft II Arena benchmark for evaluating large language models (LLMs). The table highlights three key dimensions: Strategic Planning, Real-time Decision-Making, and Adaptability. Each dimension is associated with specific metrics, such as Resource Management Ability (RMA), Resource Utilization Efficiency (RUE), Actions Per Minute (APM), and Win Rate Growth Rate (WRGR), among others. Additionally, the table details the scene selection strategies, including opponent strategies like Macro, Rush, and Random, along with the operational modes categorized as Async or Sync. This comprehensive structure facilitates a detailed assessment of LLM capabilities within complex strategic environments.

# 4 STARCRAFT II ARENA - OVERVIEW

StarCraft II Arena is a benchmark specifically designed to assess the performance of various LLMs in the strategic real-time game Starcraft II. It evaluates the capabilities of LLMs from the perspectives of strategic planning, real-time decision-making, and adaptability through a series of carefully constructed gaming scenarios. In contrast to traditional evaluation methods, StarCraft II Arena offers more refined quantitative analysis metrics and an additional behavior-tracking mechanism, allowing for a deeper, multi-faceted understanding of the underlying reasoning process of LLMs. We shall explain the detailed capability dimensions, the fine-grained capability metrics, the design of different testing scenarios, and the behavioral analysis using decision tracking as follows. Table 2

## 4.1 DECOMPOSITION OF THE STRATEGIC REASONING CAPABILITY

LLM agents demonstrate several advanced cognitive abilities akin to human intelligence in complex environments, including chain-of-thought reasoning(Wei et al., 2022; Jin & Lu, 2023; Zhang et al., 2023), planning(Huang et al., 2024a), self-reflection(Madaan et al., 2024), memory(Zheng et al., 2023a; Zhang et al., 2024b), and learning(Zhang et al., 2024a; Xi et al., 2024). Based on these characteristics, the selection of strategic planning, real-time decision-making, and adaptability as core evaluation dimensions is logically grounded. These three dimensions encapsulate the essential capabilities required for agents to tackle complex tasks, representing holistic thinking, rapid response, and flexible adaptation.

**Strategic planning** serves as the foundation for LLMs when addressing long-term objectives in dynamic environments. Short-term reactions alone are insufficient to manage fluctuating conditions. The model must have a broad view, ensuring the efficient allocation of resources, the prioritisation of tasks, and the formulation of long-term strategies to maintain a competitive edge. Effective strategic planning demands not only the ability to foresee potential future developments but also to make informed decisions concerning resource management, technological advancements, and unit production, thereby securing and sustaining a strategic advantage. In games like StarCraft II, for instance, an LLM must efficiently manage early resource accumulation and expansion while preparing for large-scale combat in the mid to late game.

**Real-time decision-making** is critical when using LLMs in real applications. While several LLM agents claim to be capable for complex tasks, some are evaluated in a setup where the executing testing system needs to be suspended while the LLM agents perform inference. A perfect strategic plan would be ineffective if the model takes too long to perform inference and cannot respond rapidly enough to the changing conditions. The model must continuously process dynamic information and adjust its tactics accordingly. For instance, during a sudden enemy assault, the LLM must promptly deploy units to defend or counter-attack, maintaining control over the situation. This capability requires not only rapid information processing but also the ability to evaluate multiple strategies quickly and efficiently to preserve overarching strategic objectives.

**Adaptability** determines the model's ability to remain competitive in evolving environments. As opponent strategies, resource conditions, and task priorities change, an adaptive model can adjust its approach based on previous feedback, refining its strategies to address new challenges. This reflects the model's flexibility and its ability to learn from experience. Quantitative metrics like win-rate growth and error-rate reduction measure how well the model improves its decision-making over time, ensuring a sustained advantage in long-term gameplay.

## 4.2 EVALUATION METRICS FOR THE INDIVIDUAL CAPABILITIES

Recent studies have highlighted that using success rate as the primary metric for agent evaluation fails to capture the nuanced differences in how language model agents perform partial tasks (Liu et al., 2023a; Li et al., 2023a). In adversarial games such as StarCraft II, this approach does not distinguish between the success of local tactics and the failure to achieve overall victory, instead treating all instances of not reaching the final objective as failures. This overlooks the agent's incremental achievements or the effectiveness of its local strategies during gameplay. Although alternative metrics like reward scores can be used to assess performance, the lack of standardisation complicates cross-environment comparisons, limiting their broader applicability(Chevalier-Boisvert et al., 2018; Wang et al., 2022; Hausknecht et al., 2020).

To address these issues, we introduce Fine-Grained Capability Metrics to provide a more precise evaluation of LLM performance across different task stages. Quantitative metrics are used to evaluate specific competencies such as resource management, real-time decision-making, and adaptability. For instance, the Resource Management Ability (RPM) is calculated by summing the total minerals and vespene gas collected during the game, reflecting the model's efficiency in resource gathering. The formula is:

$$RPM_i = \sum_{t=1}^{T}(collected\_minerals_i(t) + collected\_vespene_i(t)) \tag{3}$$

Similarly, the Supply Utilization Rate (SUR) measures the ratio of supply used to maximum supply capacity, offering insight into the model's ability to effectively produce units. The formula is:

$$SUR_i = \frac{\sum_{t=1}^{T} supply\_used_i(t)}{\sum_{t=1}^{T} supply\_cap_i(t)} \tag{4}$$

These metrics capture the model's performance across resource allocation, unit production, and technological development, providing a quantitative basis for evaluating strategic planning capabilities. A full list of these quantitative metrics, along with their respective formulas, will be provided in the appendix for reference.

## 4.3 DECISION TRACKING AND BEHAVIORAL ANALYSIS

To gain a deeper understanding of the decision-making processes and behavioural patterns of LLMs in the StarCraft II Arena, this paper introduces a decision tracking and behavioural analysis system. This system records and analyzes critical operations and decisions made by the LLM during gameplay, contextualizing them within the current game state and mission objectives to uncover the underlying strategic logic.

Specifically, the system captures key actions such as resource allocation, technological upgrades, unit production, and troop movements. A decision trace of an LLM consists of a chronological record of these key actions and decisions, detailing the context in which they were made. It typically comprises three main components: (1) Action Type, indicating the specific type of decision made (e.g., resource allocation, unit production); (2) Decision Context, which includes the game state and mission objectives at the time of the decision; and (3) Outcome, reflecting the consequences of the decision on the gameplay.

For example, a decision trace might document an LLM's choice to produce a specific unit type in response to an opponent's strategy, including the unit produced, the rationale based on the game state, and the subsequent impact on the match outcome. Screenshots illustrating these decision

| Stage | Trajectory |
|---|---|
| Early | **Decision:** 0: <TRAIN PROBE>→1: <BUILD PYLON>→2: <BUILD NEXUS>→3: <TRAIN PROBE>→4: <BUILD ASSIMILATOR> **Our Strategy:** Expand the number of workers, build additional resource collection structures, and ensure base defense. **Suggestions:** Focus on building additional Probes to boost resource gathering, construct more Pylons to increase supply cap, and consider expanding to new resource locations. `Step:429` |
| | **Decision:** 0: <TRAIN STALKER>→1: <BUILD GATEWAY>→2: <TRAIN PROBE>→3: <BUILD ASSIMILATOR> →4: <RESEARCH PROTOSSGROUNDARMORSLEVEL1> **Our Strategy:** Build a stable economy and gradually advance technology and the army. **Suggestions:** Continue to build up the economy while gradually increasing army supply and advancing technology to prepare for mid-game engagements. `Step:4629` |
| | **Decision:** 0: <BUILD GATEWAY>→1: <BUILD SHIELDBATTERY>→2: <TRAIN SENTRY>→3: <RESEARCH PROTOSSGROUNDAR-MORSLEVEL2>→4: <SCOUTING PROBE> **Our Strategy:** Build a solid economic foundation, maintain a basic military presence, and invest in early upgrades to enhance force capabilities. **Suggestions:** Consider diversifying our army composition by introducing more advanced units like Sentries or Adepts to complement the existing Zealots and Stalkers. Focus on scouting to gather intelligence on the enemy's strategy. `Step:7004` |
| Mid | **Decision:** 0: <TRAIN SENTRY>→1: <BUILD ROBOTICSFACILITY>→2: <TRAIN IMMORTAL>→3: <BUILD ASSIMILATOR>→ 4: <RESEARCH PROTOSSGROUNDARMORSLEVEL2> **Our Strategy:** Improve the effectiveness of the army around balanced economy, unit composition and early upgrades. **Suggestions:** Consider diversifying your unit composition to include more advanced units like Sentries or Immortals to counter potential enemy threats. Additionally, continue upgrading your technology to stay ahead in battles. `Step:8454` |
| | **Decision:** 0: <TRAIN IMMORTAL>→1: <TRAIN OBSERVERtgreater→2: <TRAIN VOIDRAY>→3: <BUILD GATEWAY>→4: <BUILD ASSIMILATOR> **Our Strategy:** Focus on a mix of ground and air units, supported by upgrades. We are also investing in infrastructure for advanced unit production. **Suggestions:** Consider diversifying our army composition by training units from the Robotics Facility and Stargates to counter a wider range of enemy units. `Step:9554` |
| | **Decision:** 0: <TRAIN HIGHTEMPLAR>→1: <BUILD ROBOTICSBAY>→2: <RESEARCH EXTENDEDTHERMALLANCE>→3: <BUILD ASSIMILATOR>→4: <TRAIN OBSERVER> **Our Strategy:** Focus on upgrading units and technology while maintaining a balanced economy and military composition. **Suggestions:** Consider expanding unit diversity by incorporating more advanced units like Colossi or High Templars to enhance our army's capabilities. Additionally, focus on tech upgrades to stay ahead in battles. `Step:10529` |
| Mid to Late | **Decision:** 0: <BUILD GATEWAY>→1: <BUILD PHOTONCANNON>→2: <TRAIN STALKER>→3: <RESEARCH PROTOSSAIRARMORSLEVEL2>→4: <EXPAND TO NEW RESOURCE LOCATION> **Our Strategy:** Around the mix of gateway units and technical units such as Stalkers, Immorals, and Voidray. **Suggestions:** Enhance our unit composition by adding more production structures like Gateways to increase unit production capacity. Consider expanding tech options to counter potential enemy strategies effectively. `Step:14129` |
| | **Decision:** 0: <TRAIN STALKER>→1: <BUILD GATEWAY>→2: <BUILD SHIELDBATTERY>→3: <RESEARCH PROTOSSGROUNDARMORSLEVEL2>→4: <SCOUTING OBSERVER> **Our Strategy:** Around the unit diversity and balance method of technological progress. The emphasis on stalkers suggests a flexible military composition capable of handling a variety of threats. **Suggestions:** Consider diversifying our unit composition further to adapt to potential enemy strategies. Focus on maintaining map control and scouting to stay ahead of the opponent. `Step:17554` |
| | **Decision:** 0: <TRAIN STALKER>→1: <BUILD GATEWAY>→2: <CHRONOBOOST CYBERNETICSCORE>→3: <RESEARCH PROTOSSAIRWEAPONS-LEVEL2>→4: <TRAIN IMMORTAL> **Our Strategy:** Focus on balanced unit composition and technology upgrades to improve the effectiveness of our forces. **Suggestions:** Consider diversifying our army composition further to counter the enemy's Stalkers effectively. Focus on unit production and upgrades to strengthen our army. `Step:20929` |

Table 3: Presents a detailed decision trajectory analysis of an LLM's gameplay in StarCraft II across various stages: Early, Mid, and Mid to Late. The table outlines specific decisions made by the model, including the training of units, building structures, and conducting research. Each entry details the decision-making process, the associated strategy, and suggestions for optimizing performance. By capturing these trajectories, the table illustrates how the LLM navigates complex strategic choices, adapts to the game environment, and develops its military and economic strategies over time, providing insights into its strategic reasoning capabilities.

traces will be provided in the appendix to offer visual clarity.These actions are traced throughout the entire decision chain to assess the coherence of the LLM's strategic planning, flexibility in tactical adjustments, and adaptability to opponent strategies. Additionally, the system employs visualization tools to present real-time behavioural pathways, allowing researchers to observe how the model reacts in various scenarios. This real-time tracking provides insights into effective decision-making patterns and potential areas for optimization, offering a comprehensive understanding of both tactical execution and strategic intent.

In parallel, qualitative metrics are introduced to further analyze the LLM's decision-making patterns and strategic innovations. These metrics are closely tied to the decision tracking system, capturing specific game actions such as the Unit Construction Order, which reflects the model's tactical priorities by documenting the sequence in which units are built. For example, prioritizing basic units may indicate a focus on early offensive strategies, whereas building high-tech units suggests a defensive or late-game approach. Similarly, the Key Building Completion Time records the timing of critical structures, such as Nexus or Gateway, to determine whether the construction order supports economic growth or military objectives.Moreover, the Strategy Innovation Rate, which measures the frequency of adopting new strategies across multiple games, is derived from continuous monitoring of strategic shifts. These qualitative metrics provide a detailed view of the model's adaptability and capacity for innovation across various game scenarios. Definitions and methods for these metrics will be further detailed in the figure4.3, along with examples and screenshots to effectively illustrate the decision tracking process.

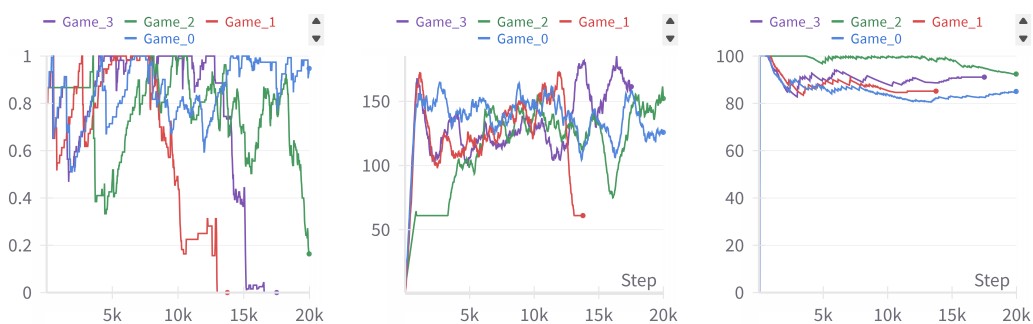

(a) Strategic Planning - EER (b) Real-time Decision Making - EPM (c) Adaptability - ERT

Figure 2: Performance indicators for evaluating LLM capabilities in StarCraft II: (a) Strategic Planning - EER (Efficiency of Resource Utilization), (b) Real-time Decision Making - EPM (Effective Actions Per Minute), and (c) Adaptability - ERT (Error Rate Trend). Each graph displays the performance trends of different game sessions ($Game_0$, $Game_1$, $Game_2$, $Game_3$) over time steps.

# 5 ARENA EVALUATION

## 5.1 EVALUATION SETUP

We conducted a comprehensive evaluation of popular large language models, including both proprietary API-based models and open heavyweight models. Firstly, we report the success rates and progress rates of these agents. Subsequently, we provide a detailed analysis of their performance and measure the various capabilities of the LLM agents, culminating in a further analysis of their decision-making trajectories.

## 5.2 MAIN RESULTS

| Model | Win Rate | Strategic Planning | Real-Time Decision | Adaptability | Overall Score |
|---|---|---|---|---|---|
| GPT-4o(OpenAI, 2024b) | 2/10 | 62.01 | 21.12 | 38.64 | 57758 |
| GPT-4o mini(OpenAI, 2024a) | 5/10 | 54.71 | 37.51 | 24.52 | 62541 |
| GPT-3.5 Turbo(OpenAI, 2023) | 4/10 | 53.24 | 36.23 | 47.41 | 60914 |
| Gemini 1.5 Flash(Reid et al., 2024) | 5/10 | 37.56 | 39.34 | 38.18 | 55940 |
| DeepSeek-V2.5(DeepSeek-AI, 2024) | 2/10 | 58.35 | 37.11 | 20.16 | 43070 |
| Llama-3.1-8B-Instruct(Dubey et al., 2024) | 3/10 | 37.56 | 47.05 | 32.71 | 44901 |
| Llama-3.1-70B-Instruct(Dubey et al., 2024) | 2/10 | 55.44 | 30.24 | 52.77 | 46825 |

Table 4: Demonstrates the performance of several large-scale language models on different ability dimensions, specifically win rate, strategic planning, social reasoning, real-time decision making, teamwork, learning ability, and overall score.

**Fine-grained capability metrics provide a more detailed and insightful evaluation of model performance than simple success rates.** These metrics reveal substantial differences in how models handle strategic planning, real-time decision-making, and adaptability. GPT-4o achieved the highest score in strategic planning with 62.01 points, showcasing its strength in long-term resource management and strategy. However, its real-time decision-making score of 21.12 points was notably lower, indicating slower response times to in-game events. Conversely, Llama 3.1 Instruct 8B excelled in real-time decision-making with a score of 47.05, yet its strategic planning score was lower at 37.56, suggesting it is better suited to making quick decisions under pressure rather than managing long-term strategies. Llama 3.1 Instruct 70B led in adaptability, particularly in metrics like win rate growth and error rate reduction, which reflects its ability to learn and adjust to evolving game conditions. These findings demonstrate that fine-grained metrics enable a more nuanced understanding of each model's strengths and weaknesses, beyond what win rates alone can offer.

**Evaluating performance in both synchronous and asynchronous settings reveals how time constraints impact model behaviour.** In synchronous settings, where rapid decision-making is essential, Llama 3.1 Instruct 8B and GPT-4o mini performed exceptionally well, with real-time decision-making scores of 47.05 and 37.51, respectively, highlighting their ability to respond quickly to changing conditions. However, in asynchronous settings, where models have more time to process information and plan their strategies, GPT-4o and DeepSeek-V2.5 excelled, achieving strategic planning scores of 62.01 and 58.35, respectively. This contrast illustrates that models adept at quick decision-making may face challenges in handling complex, long-term planning, while those strong in strategic planning may be slower to react in time-critical situations. Therefore, considering both settings is crucial for a comprehensive evaluation, as it underscores the balance between short-term reactivity and long-term planning in model performance.

**Closed-source models consistently outperform open-source models in strategic planning and overall performance.** GPT-4o and GPT-4o mini were the top performers, with GPT-4o achieving the highest overall score of 57,758, significantly surpassing open-source models such as Llama 3.1 Instruct 70B and Llama 3.1 Instruct 8B, which scored 46,825 and 44,901, respectively. This demonstrates that closed-source models benefit from larger training datasets and more optimised architectures, giving them an advantage in resource management and strategic tasks. Despite this, open-source models like Llama 3.1 Instruct 8B showed competitive performance in real-time decision-making, scoring 47.05, suggesting that open models are better suited for tasks requiring rapid responses. While closed-source models dominate in long-term planning and complex reasoning, the performance of open-source models in real-time decision-making highlights their potential, especially with further optimisation and development. This suggests that with additional resources, open-source models could narrow the performance gap, particularly in more complex strategic tasks.

## 5.3 ANALYTICAL EVALUATION

**Unit Construction Order.** The Unit Construction Order is a critical metric that reflects the tactical priorities of the LLM. As illustrated in Table 4.3, the decision trace reveals the sequence of units constructed during gameplay, allowing us to assess strategic intent. For instance, in Game 3, a notable shift occurs as the game progresses into the later stages, with the LLM beginning to prioritize the construction of advanced units. This transition can be observed through the timing of unit production, which indicates that the model is adapting its strategy in response to the evolving game dynamics. For example, the LLM initially focuses on building basic units, which is typical in the early game to establish a strong economy and military presence. However, as the game advances, there is a marked increase in the production of higher-tier units, such as Colossi and High Templars. This suggests a strategic shift aimed at countering opponent threats and enhancing combat effectiveness. The visual representation in Figure 2 further emphasizes this point, showing the timing and frequency of unit production across different game sessions.

**Key Building Completion Time.** The Key Building Completion Time metric assesses the efficiency and timing of critical structures necessary for advancing the game's strategy. In Game 3, we observe that the completion of vital buildings such as the Robotics Facility and Templar Archives coincides with the shift towards producing more advanced units. This timing indicates that the LLM is effectively managing its resources to maximize its strategic output. For instance, if the completion time for these structures is relatively short and aligns with the LLM's decision to construct advanced units, it reflects a well-coordinated strategy that prioritizes technological advancement alongside unit production. This synchronization is crucial for maintaining pressure on opponents and capitalizing on strategic opportunities.

**Strategy Innovation Rate.** The Strategy Innovation Rate measures the frequency with which the LLM adopts new strategies during gameplay. By analyzing the decision traces, we can identify instances where the model implements novel tactics or unit combinations in response to evolving game conditions. For example, in the later stages of Game 3, the LLM demonstrates an increase in strategic innovation, as evidenced by its willingness to experiment with unit compositions that differ from those used in earlier phases. This adaptability is highlighted in Figure 2, where we can see fluctuations in performance metrics over time. Such fluctuations suggest that the LLM is actively refining its strategies to better respond to opponents and the overall game state. This capacity for innovation is a testament to the model's robust decision-making framework, enabling it to remain competitive in a dynamic environment.

## 6 CONCLUSION

This study presents StarCraft II Arena as a comprehensive benchmark for assessing the capabilities of LLMs in strategic planning, real-time decision-making, and adaptability. The findings demonstrate that LLMs possess varying strengths across these dimensions, with notable performance in strategic reasoning and adaptability. By employing fine-grained metrics, we highlight the limitations of traditional success rates in capturing the true decision-making processes of LLMs. Our analysis reveals that models like GPT-4o excel in long-term strategic planning, while others, such as Llama 3.1 Instruct 8B, exhibit superior real-time decision-making capabilities. This detailed evaluation not only enhances our understanding of LLMs' cognitive abilities in complex environments but also lays the groundwork for future advancements in AI research, emphasizing the importance of dynamic assessments in evaluating AI agents.

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

# A APPENDIX

## A.1 EVALUATION METRICS

| Capacity | Computer Formula |
|---|---|
| Strategic planning | $\text{RPM}_i = \sum_{t=1}^{T} \left( \text{collected\_minerals}_i(t) + \text{collected\_vespene}_i(t) \right)$ 

 $\text{EER}_i = \frac{\sum_{t=1}^{T} \left( \text{collected\_minerals}_i(t) + \text{spent\_vespene}_i(t) \right) \times 100}{\sum_{t=1}^{T} \left( \text{collected\_minerals}_i(t) + \text{collected\_vespene}_i(t) \right)}$ 

 $\text{SUR}_i = \frac{\sum_{t=1}^{T} \text{supply\_used}_i(t)}{\sum_{t=1}^{T} \text{supply\_cap}_i(t)}$ 

 $\text{TRR}_i = \frac{\text{completed\_tech}_i}{\text{total\_research\_count}_i}$ |
| Real-time decision making | $\text{APM}_i = \frac{\text{total\_actions}_i}{\text{game\_time\_minutes}_i}$ 

 $\text{EPM}_i = \frac{\text{effective\_actions}_i}{\text{game\_time\_minutes}_i}$ |
| Adaptability | $\text{WinRateTrend}_i = \frac{\text{WinRate}_{\text{end,i}} - \text{WinRate}_{\text{start,i}}}{\text{total\_games}}$ 

 $\text{ErrorRateTrend}_i = \frac{\text{ErrorRate}_{\text{start,i}} - \text{ErrorRate}_{\text{end,i}}}{\text{total\_games}}$ |

Table 5: This table presents the key capacity metrics and their corresponding computational formulas used to evaluate LLMs in StarCraft II. The metrics are categorized under three primary capacities: Strategic Planning, Real-time Decision Making, and Adaptability. Each metric captures different aspects of the LLM's performance, such as resource management (RPM), supply utilization (SUR), action efficiency (APM, EPM), and adaptation trends (WinRateTrend, ErrorRateTrend), providing a comprehensive assessment of the model's gameplay capabilities.

## A.2 GAMES INTRODUCTION

**StarCraft II** is a real-time strategy game whose core mechanics include resource management, base building, troop production and command. Players need to efficiently gather resources, build and upgrade bases, train various military units, and defeat opponents through precise micromanagement and macro-strategy in a real-time environment. The game emphasizes quick decision-making and flexibility, requiring players to balance economic development and military operations in a highly dynamic battlefield in order to ultimately destroy the enemy's base and win.

The core mechanics of Civilization revolve around turn-based strategy, where players lead a civilization from antiquity to the future by managing cities, developing technology and culture, exploring maps, and engaging in diplomacy and warfare. With an emphasis on resource management, long-term planning, and strategic decision-making, the game requires players to unlock new abilities through the tech and culture trees, choose different victory conditions (e.g., military victory, tech victory, or cultural victory), and gain an advantage in their interactions with other civilizations. The variety and depth of the game makes it a classic strategy game.

| Game manual | | |
|---|---|---|
| **Setting** | Map specification | Standard 1v1 map with mining, gas, expansion points, obstacle terrain and other elements (e.g. map: Jagannatha LE). |
| | Number of players | 2 players per match against each other. |
| | Resource type | Two main resources - minerals and gases, used for unit production and technological upgrading. |
| **Unit configuration and policy** | Basic unit configuration | 12 farmers (SCV/Probe/Drone) for resource collection. |
| | | 1 main base (Command Center/Nexus/Hatchery). |
| | | 1 Supply Depot (Pylon/Overlord) to control the population cap. |
| | Ethnic divisions | Terran: Focuses on mechanical units and air power, with strong defensive and multi-functional building capabilities. |
| | | Protoss: has shields and powerful individual units, but is slower to produce. |
| | | Zerg: Unit production is fast, relying on massive ground forces and good ecological control. |
| | Unit Production and Technology tree | Terran: can produce ground units (such as Marine, Marauder) and air force units (such as Viking, Banshee). |
| | | Protoss: Can produce high-attack units (e.g., Zealot, Stalker) and powerful air units (e.g., Carrier, Phoenix). |
| | | Zerg: Can produce a large number of cheap units (such as Zergling, Hydralisk) and high-tech units (such as Mutalisk, Ultralisk). |
| **Fixed opening strategy** | Initial base strategy | Rapid expansion strategy: quickly establish a second base to enhance economic output and increase resource collection speed. |
| | | Quick attack strategy: Quickly produce early combat units, directly attack enemy bases, forcing opponents to defend. |
| | | Defensive strategy: Strengthen fortifications (such as Terran's Bunker, Protoss 'Photon Cannon) to delay enemy attacks and save strength for later development. |
| | Army layout and defense | Defensive arrangement: Arrange defensive units near the base to ensure the safety of the mining area and the main base; Different races have different defensive structures, such as Terran's Bunker, Protoss 'Shield Battery, and Zerg's Spine Crawler. |
| | | Offensive placement: Deploy units to harass and control key locations on the map, such as enemy resource points. |
| **Goals** | Economic development | Through the collection of minerals and gases, the rapid development of the economy and science and technology, the establishment of a more powerful army. |
| | Military victory | Destroy all their main bases or render them incapable of reproducing units. |
| | Map control | Capture key positions on the map (e.g., resource points, highlands) and use tactical advantage to overwhelm the opponent's economy. |

Table 6: Game manual detailing the settings, unit configuration and policy, fixed opening strategies, and goals for gameplay in StarCraft II. This table provides a comprehensive overview of the game mechanics, including map specifications, player setup, unit production capabilities, strategic approaches, and objectives essential for effective gameplay.

