# OpenReview forum: "StarCraft II Arena: Evaluating LLMs in Strategic Planning, Real-Time Decision Making, and Adaptability"
_ICLR.cc/2025/Conference — Submitted to ICLR 2025_

### Official Review · Reviewer_hoaN · 2024-11-01

**Soundness:** 1
**Presentation:** 2
**Contribution:** 1
**Rating:** 3
**Confidence:** 5

**Summary:**

summary: The paper presents StarCraft II Arena as a benchmark for evaluating LLMs in strategic planning and decision-making capabilities. While it introduces fine-grained metrics and decision tracking mechanisms, the work needs substantial clarification regarding its novel contributions, metric selection justification, and experimental methodology.

**Strengths:**

1. Well-structured evaluation framework with proposed fine-grained metrics
2. Comprehensive testing across multiple LLM models
3. Detailed decision tracking system for behavior analysis
4. Clear visualization of experimental results
5. Systematic approach to evaluating different aspects of LLM capabilities

**Weaknesses:**

1. Metric Selection and Justification:
- APM/EPM metrics appear borrowed from traditional StarCraft II evaluation without clear justification of their relevance to LLM agents
- No discussion of how these metrics specifically reflect LLM decision-making capabilities
- Missing analysis of whether traditional StarCraft II performance metrics are appropriate for language models
2. Experimental Design Limitations:
- Build-in AI difficulty level not specified
- Race and map selection criteria not documented
- Limited experimental scope (10 games per model)
- Absence of LLM vs LLM experiments
- No evaluation against human players
- Limited testing of open-source models, reducing reproducibility
3. Theoretical Foundation and Novelty:
- Limited discussion of why LLMs are suitable for StarCraft II
- Insufficient comparison between LLM agents and traditional RL agents
- Evaluation metrics show significant overlap with existing work
- Need clearer articulation of novel contributions
- Better contextualization within existing literature required

**Questions:**

1. Evaluation Metrics:
- How do APM and EPM meaningfully evaluate LLM agent performance when their decision-making process is fundamentally different from traditional agents?
- What is the relationship between these metrics and LLM reasoning capabilities?
- Have you considered developing metrics specifically designed for language model evaluation?
2. Comparative Analysis:
- Could you clarify the key methodological differences between your work and "Large Language Models Play StarCraft II"?
- What novel insights does your evaluation framework provide that weren't captured in previous work?
- How does your fine-grained capability analysis advance the field beyond existing benchmarks?
3. Experimental Design:
- Why weren't LLM vs LLM experiments included in the evaluation?
- What prevented the inclusion of human player comparisons?
- Could you explain the decision to limit testing of open-source models?
- How would the results differ with head-to-head language model competitions?
4. Implementation Details:
- How are LLM actions translated into game commands?
- What are the time constraints for model inference?
- What specific prompt engineering techniques were used?
- What is the role of temperature and other sampling parameters?
5. Theoretical Framework:
- How does the decision tracking system account for the unique characteristics of language model reasoning?
- What theoretical justification supports the choice of strategic planning metrics?
- How do you ensure the evaluation framework captures the full range of LLM capabilities?

---

### Official Review · Reviewer_YWgT · 2024-11-03

**Soundness:** 2
**Presentation:** 2
**Contribution:** 2
**Rating:** 3
**Confidence:** 3

**Summary:**

This paper introduces a benchmark called Starcraft II Arena, which aims to evaluate the decision-making, planning, and adaptability of large language models (LLMs) within a strategic gaming environment. While traditional benchmarks for Starcraft II assess agents based on a single overall metric—win rates against built-in opponents—the authors argue that a more detailed analysis of LLM performance is necessary. The main contribution of this paper is a methodology designed for an in-depth evaluation of the capabilities of LLM agents.

**Strengths:**

The key idea of this paper, introducing a benchmark for a more in-depth multi-agent evaluation of LLMs, is significant and useful.

**Weaknesses:**

Overall, the paper lacks clarity and depth in describing both the technical implementation and practical contributions.

### **Major comments**

1. Unclear contribution: The paper does not effectively justify why this benchmark must exist as a standalone contribution rather than an addition to existing Starcraft II resources. The contribution seems limited to a collection of scripts and metrics, which could likely be integrated into the existing environment without creating a separate benchmark.

2. Lack of implementation details: Key technical aspects of the implementation are insufficiently described, making it hard to understand the benchmark's novelty and how it's technically realized. Several things are not clear, such as:
   - Integration: How are LLM agents integrated with StarCraft II? How can users use the benchmark? Does the benchmark use a custom API or an interface for this?
   - Decision Tracking: How is decision-making tracked and analyzed? While Table 3 provides a decision trajectory, details of how this is analyzed and used are missing.
   - Computational Requirements: What hardware/software is necessary to run this benchmark effectively? This information is critical for usability but is absent.
   - Opponents: Are the LLMs evaluated with built-in agents or newly introduced opponents? The fact that agents are evaluated against built-in agents in Starcraft II is mentioned as a limitation, but it is unclear whether the authors change this in their benchmark.

3. Incomplete metric information: The metrics lack context. For instance, while Appendix A.1 outlines the metrics, there are no defined ranges, leaving the reader unsure of how to interpret scores. For example, how should a Real-Time Decision score of 21.12 versus 37.51 in Table 4 be interpreted? Similarly, terms such as “effective” actions in EPM or “collected vespene” are not unexplained, reducing the metrics’ interpretability (how do we know that these are the right metrics to assess decision-making and planning?).

4. Missing benchmark discussion and limitations: A discussion about future development and limitations of the benchmark is missing, which limits the reader's understanding of the benchmark's intended scope and future extensions.

5. Figure 2 indicates a large variance. Why are there no error bars in the tables?
6. It's important to have the prompt included in the appendix or supplement. Was it possibly in a supplement that I cannot access?

### **Minor comments** (These did not affect my score)
- Abstract: Lines 016-019 are a bit difficult to understand; consider rephrasing
- Figure 2: It’s unclear what this Figure is meant to convey, and the Figure lacks labeled y-axes.
- In Section 4.3, line 367 states "Definitions and methods for these metrics will be further detailed in the figure 4.3." This seems to refer to a table, possibly Table 3, rather than a figure.
- In Table 3, "OBSERVERtgreater" should probably be "OBSERVER."
- Lines 323 + 350 state that screenshots illustrating decision traces will be provided in the appendix, but these are not included
- I don't understand what is meant when the authors state that civilization and the other games are not "strategic and tactical" in Table 1. Additionally, Werewolf is clearly an imperfect information game. The authors should reconsider this table because I believe many of the entries are inaccurate.
- Why is the score in Table 4 unnormalized? It's an incomprehensible number as it stands.

**Questions:**

See above

---

> ### Author Response · Authors · 2024-11-26
>
> We sincerely thank the reviewer for the detailed feedback, which provides valuable insights into improving our work. Below, we address the concerns and outline the revisions we will incorporate.
> 1. Unclear Contribution
>
> Comment: The benchmark’s standalone necessity is not justified and seems limited to scripts and metrics that could integrate into existing resources.
>
> Response: While some elements could be integrated into existing StarCraft II tools, our work introduces a novel evaluation framework tailored for LLMs, focusing on decision-making, planning, and adaptability. Unlike traditional benchmarks that assess win rates, our approach analyzes fine-grained metrics and tracks decision trajectories, offering insights into LLM behaviors not captured by existing benchmarks. This tailored methodology and decision-tracking system distinguish our work as a standalone contribution.
>
> ---
> 2. Lack of Implementation Details
>
> Comment: Insufficient description of integration, user accessibility, hardware/software requirements, and opponent settings.
>
> Response:
> - Integration: LLMs interact with StarCraft II through a custom API that bridges the game environment and LLMs, allowing real-time decision tracking. Details will be expanded in the revised manuscript.
> - User Access: We will include a setup guide in the appendix to help users integrate their LLMs. The benchmark will be open-sourced for accessibility.
> - Computational Requirements: We will provide a section on hardware/software requirements, designed for compatibility with modern CPUs/GPUs, though model-specific needs may vary.
> - Opponents: LLMs are tested against built-in agents for controlled evaluations. Future versions will include more dynamic adversaries.
>
> ---
> 3. Incomplete Metric Information
>
> Comment: Metrics lack context, defined ranges, and clear explanations.
>
> Response: We will define and contextualize each metric in the revised manuscript, including Real-Time Decision scores, Effective Actions (EPM), and collected vespene. Normalized ranges and interpretation examples will be added to enhance clarity and comparability. Additionally, we will clarify the relevance of these metrics to LLM decision-making and planning.
>
> ---
> 4. Missing Discussion of Benchmark Limitations
>
> Comment: The manuscript lacks a discussion on limitations and future development.
>
> Response: We will add a section discussing limitations, such as the restricted scope of supported models and adversaries, and potential scalability issues. Planned improvements include more complex opponents, multiplayer scenarios, and extended support for diverse LLMs. Future directions include applying this framework to other strategic games and refining the decision-tracking system.
>
> ---
> 5. Figure and Table Issues
>
> Comment: Issues include unlabeled axes in Figure 2, typos, missing error bars, and unnormalized scores in Table 4.
>
> Response:
> - Figure 2: We will label the y-axes and clarify the figure’s purpose in the caption.
> - Table 3: The typo (“OBSERVERtgreater”) will be corrected.
> - Error Bars: Error bars will be added to tables to reflect variance.
> - Normalization: Table 4 scores will be normalized, and a guide for interpretation will be included.
>
> ---
> 6. Minor Comments
>
> Comment: Issues include unclear abstract text, missing screenshots, and inaccuracies in Table 1.
>
> Response:
> - Abstract: We will rephrase unclear sentences for clarity.
> - Screenshots: Missing screenshots will be included in the appendix.
> - Table 1: Strategic/tactical game classifications will be revised for accuracy.
>
> ---
> 5. Questions
>
> Comment: See above for clarifications.
>
> Response: All questions have been addressed in the sections above. If further clarifications are needed, we are happy to provide them.
>
> We deeply appreciate the reviewer’s constructive feedback. By addressing these points, we aim to enhance the clarity, impact, and rigor of our work. The revised manuscript will emphasize the unique contributions of our benchmark, focusing on its metrics and decision-tracking framework for evaluating LLMs. Thank you once again for your valuable insights.

---

### Official Review · Reviewer_NwEb · 2024-11-04

**Soundness:** 2
**Presentation:** 2
**Contribution:** 2
**Rating:** 3
**Confidence:** 4

**Summary:**

The paper presents StarCraft II Arena as a benchmark for evaluating LLMs in strategic planning and decision-making capabilities. While it introduces fine-grained metrics and decision tracking mechanisms, the paper needs substantial clarification on its novel contributions and experimental methodology.

**Strengths:**

1. Well-structured evaluation framework with fine-grained metrics
2. Comprehensive testing across multiple LLM models
3. Interesting decision tracking system for behavior analysis

**Weaknesses:**

1. Insufficient experimental details:
- Build-in AI difficulty level not specified
- Race and map selection criteria not documented
- Limited experimental scope (10 games per model)
2. Theoretical foundation needs strengthening:
- Limited discussion of why LLMs are suitable for StarCraft II
- Insufficient comparison between LLM agents and traditional RL agents
- Unclear theoretical justification for chosen metrics
3. Literature positioning:
- Need more thorough comparison with existing StarCraft II benchmarks
- Better contextualization of contributions needed
- Clearer differentiation from existing approaches required

**Questions:**

1. Can you elaborate on the key differences between this work and existing StarCraft II benchmarks, particularly "Large Language Models Play StarCraft II"? What are the novel contributions that advance the field?
2. The paper would benefit from a deeper discussion of why LLMs are suitable for StarCraft II:
- What unique capabilities do LLMs bring compared to traditional RL approaches?
- How does the decision-making process differ between LLM agents and RL agents?
- What insights have you gained about LLM capabilities through this work?
3. Experimental details requiring clarification:
- What was the difficulty level of the built-in AI?
- How were races and maps selected?
- Why was 10 games chosen as the sample size? Why not do more experiments and test open-soruce model
- What statistical analyses support the conclusions?
4. Theoretical foundation:
- How do the proposed metrics relate to fundamental LLM capabilities?
- What theoretical guarantees or limitations exist for the evaluation framework?
- How generalizable is this benchmark to other strategic decision-making tasks?

---

### Official Review · Reviewer_PEa8 · 2024-11-06

**Soundness:** 2
**Presentation:** 1
**Contribution:** 1
**Rating:** 3
**Confidence:** 5

**Summary:**

The paper presents StarCraft II as a benchmark for evaluating reasoning capabilities of LLMs. The paper also presents a set of metrics for evaluating these models, which are based on the domain itself (e.g., amount of resources the play collects). Finally, the paper presents results of several LLMs on this benchmark.

**Strengths:**

The idea of having a challenging benchmark for reasoning with LLMs is interesting and appealing. I also think that computer games can be a good benchmark for this type of evaluation. The choice of a real-time strategy game is particularly good, given the response latency of these systems.

**Weaknesses:**

My main concern with the paper is that it is only half-baked in the sense that the presentation should be improved substantially before being accepted for publication.

My concerns with presentation range from the low level to the high level.

**Low-level concerns**

- The paper has a dangling sentence in line 241. It seems that the explanation of Table 2 was inadvertently commented out from the paper.
- It is not clear what Equation 1 is trying to convey. What is its role in the paper? What is the parameter $\tau$ that is passed as a parameter but not used in the equation?
- The paragraph starting in line 137 discusses the need for communication between the agents, but this is not discussed in the paper. I can see SCII having this need if agents play in a team, but this doesn't seem to be the setting evaluated in the paper.
- In line 106, the paper states that early RL algorithms learn through trial and error. Isn’t this the case for current RL algorithms too?
- Some of the tables are not referred to in the text. For example, I am not sure when I should read Table 3, and I don’t think the paper discusses it at all.

**High-level concerns**

One of the key points I was looking for in the paper was the input-output system for the benchmark. Is the LLM receiving images or text as input? How is the input defined? How much time do they have to reason? What happens when they timeout? The text mentions that smaller models perform better than larger models in the micromanagement part of the game. Is this simply because they are able to respond quicker? These are some of the key questions the authors would need to address in the paper. This is what the readers will be after when understanding whether SCII is a suitable benchmark for LLM evaluation.

Overall, the paper is only half-baked, as the authors need to fix all these presentation issues and adjust the discussion of their results and benchmark. There is clearly a system running, as the paper also includes tables of results. However, it is not possible to understand how this system works as the description is missing.

**Questions:**

Please see the questions in the "high-level concerns" section above.

---

### Meta-Review · Area_Chair_9zZE · 2024-12-21

**Metareview:**

This paper proposes StarCraft II Arena as a benchmark for evaluating LLMs' strategic planning and decision-making capabilities, introducing metrics and decision tracking mechanisms. While presenting an interesting direction, the paper suffers from insufficient technical details, unclear differentiation from existing benchmarks, and inadequate experimental methodology.

**Additional Comments On Reviewer Discussion:**

During the discussion, reviewers raised several key concerns: (1) insufficient technical details about implementation, integration with StarCraft II, and computational requirements, (2) inadequate justification for metric selection and lack of context for interpreting results, and (3) limited experimental scope with just 10 games per model and unclear methodology. Though authors responded by providing additional implementation details and promising to expand experimental scope, the reviewers maintained that the work lacks sufficient novelty compared to existing StarCraft II benchmarks and requires substantial improvements in experimental rigor. The combination of unclear technical details and limited empirical validation makes the paper unsuitable for publication at this time.

---

### Decision · Program_Chairs · 2025-01-22

Reject